# Hybrid Model Development for HVAC System in Transportation

**Antonio Gálvez** [1,2,*] **, Dammika Seneviratne** [1] **and Diego Galar** [1,2]

1   TECNALIA, Basque Research and Technology Alliance (BRTA), 48170 Derio-Vizcaya, Spain;
    dammika.seneviratne@tecnalia.com (D.S.); diego.galar@tecnalia.com or diego.galar@ltu.se (D.G.)
2   Division of Operation and Maintenance Engineering, Luleå University of Technology, 971 87 Luleå, Sweden
*   Correspondence: antonio.galvez@tecnalia.com or antonio.galvez@ltu.se

**Abstract:** Hybrid models combine physics-based models and data-driven models. This combination is a useful technique to detect fault and predict the current degradation of equipment. This paper proposes a physics-based model, which will be part of a hybrid model, for a heating, ventilation, and air conditioning system installed in the passenger vehicle of a train. The physics-based model is divided into four main parts: heating subsystems, cooling subsystems, ventilation subsystems, and cabin thermal networking subsystems. These subsystems are developed when considering the sensors that are located in the real system, so the model can be linked via the acquired sensor data and virtual sensor data to improve the detectability of failure modes. Thus, the physics-based model can be synchronized with the real system to provide better simulation results. The paper also considers diagnostics and prognostics performance. First, it looks at the current situation of the maintenance strategy for the heating, ventilation, air conditioning system, and the number of failure modes that the maintenance team can detect. Second, it determines the expected improvement using hybrid modelling to maintain the system. This improvement is based on the capabilities of detecting new failure modes. The paper concludes by suggesting the future capabilities of hybrid models.

**Keywords:** hybrid modelling; digital twins; physics-based model; HVAC; transportation engineering; simulations





## 1. Introduction

Corrective maintenance is a common strategy for equipment maintenance, but companies attempt to avoid it for safety, reliability, economic, and environmental reasons. In addition, the technological improvements accompanying the fourth industrial revolution are giving companies a broader range of possibilities to improve the maintainability, availability, and reliability of their equipment. The need to avoid corrective maintenance, together with today's technical advances, means that companies are turning to predictive maintenance (PdM), implementing it as an information source to reduce maintenance costs, extend an asset's useful life, and improve the reliability and availability. To this end, they are implementing diagnostics and prognostics in condition-based monitoring (CBM) as part of their overall prognostics and health management (PHM) plan.

Given companies' interest in PdM, researchers are working on developing tools that are able to predict the current health state and estimate the remaining useful life (RUL) of components, a key requirement of PHM. The new tools and methodologies are providing feedback and, this feedback, in turn, is defining the ongoing investigation of predictive maintenance.

The system studied and modelled in this paper is a heating, ventilation, and air conditioning (HVAC) system that is used in a passenger train. This system keeps the cabin of the vehicle at a comfortable temperature, with an acceptable concentration of $CO_2$. A failure in this system directly affects people, which makes safety the paramount factor in maintenance decisions [1].

Diagnostics is the identification of a faulty component through the detection and isolation of a fault. Diagnostics processes detect a fault and identify the faulty part when it

displays a known failure mode (FM). Thus, the implementation of diagnostics includes failure mode and effects analysis (FMEA) [1,2]. Diagnostics starts once a fault or abnormal behavior is detected, but the component in an unhealthy state could cease to operate or continue to be operational in a degraded mode. In contrast, the prognostics process continuously estimates the RUL, which is an important support for health assessment.

A broad variety of models have been developed in transportation industry, such as [3,4]. Nevertheless, this research work is focused on models orientated to fault detection and the estimation of components degradation. As Diego Galar and Uday Kumar explained [2], data-driven approaches, model-based approaches, hybrid model approaches (HyMAs), and experience-based approaches are currently the main techniques used to build RUL estimation models for diagnostics and prognostics. Figure 1 shows the classifications.

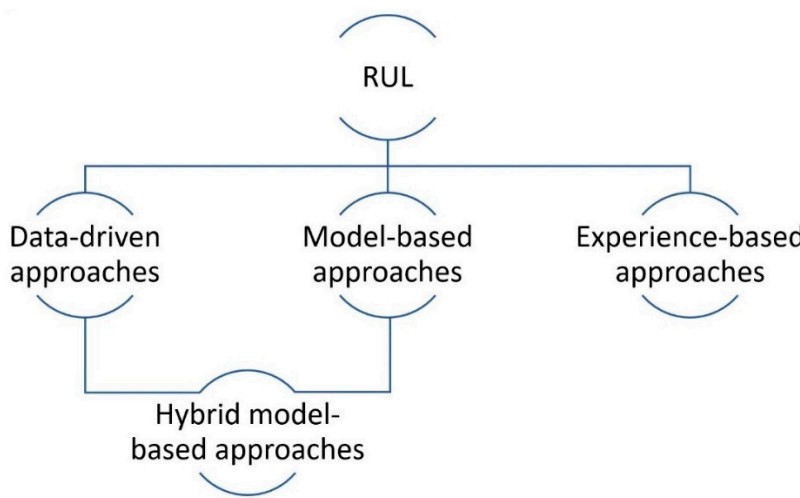

**Figure 1.** Remaining useful life (RUL) estimation models.

This research work presents a hybrid model that combines physics-based models and data-driven models. The models are developed for an HVAC system using MATLAB R2019b. The paper proceeds, as follows: Section 2 describes the technical approaches developed and the literature review related to the technical approaches. Section 3 describes the problems using data-driven approaches for this system. Section 4 explains the proposed hybrid model, their advantages, and the methodology used for combining both models. Section 5 describes the physics-based model developed and described the physics-based model for the cabin thermal networking in detail. Section 6 describes the data-driven model developed. Section 7 discusses the results that were obtained after validating and testing the hybrid model. Section 8 mentions the conclusions and outlook of this research work.

## 2. Literature Review of Technical Approaches

Physics-based model approaches are explicit mathematical models of an asset. These approaches estimate the RUL of the system by giving the model an understanding of the physics of the monitored system [5]. Sometimes they cannot be used, specifically in a complex system or process where some key parameters are very difficult or impossible to obtain, often because doing so requires too many resources. Nevertheless, an accurate physics-based model approach is more effective than other approaches [6]. Bendapudi, Braun, and Groll [7] introduced a dynamic model of an HVAC system for fault detection and diagnostics (FDD). The paper includes different cases using model-based approaches for FDD [8,9]. MATLAB Simulink is a powerful tool for building models. The literature of this research work contains detailed physics-based models of HVAC systems installed in different assets [10–12].

Data-driven model approaches are used for fault detection and the estimation of the RUL by analyzing the data acquired from the system. The approach gives results for

diagnostics and prognostics by building a model from data that were directly taken from the system. Refs [13,14] presented the use of approaches based on data for fault diagnosis of HVAC chillers. More recently, data-driven methods were used for fault detection and diagnostics in air handling units by Montazery and Kargar [15]. In another recent paper, Zhou and Zhenzin compared several basic data-driven fault diagnostics methods for a system with variable refrigerant flow [16].

A hybrid model approach (HyMA) combines information from models built based on data and knowledge from the model based on physical laws in order to improve diagnostics and prognostics capabilities, as shown in Figure 1. Nevertheless, researchers use the term HyMA in different ways [6]. There are some researchers who use the term hybrid modelling for the combination of data-driven techniques [17–19]. The literature of this research work contains interesting researches that combine the physics-based model, data-driven model, and experience-based model for fault detection [20–25].

The maintainers of the HVAC system under study are currently using approaches that are based on data, but they must use preventive maintenance in critical components because of the insufficient historical data for training the prediction model. The physics-based model that is mentioned here is used to overcome the lack of data. This also reduces the number of failure modes (FMs) that are hidden and have not yet occurred, which are metaphorically known as "black swan losses" [26]. Moreover, the hybrid model id developed to estimate the obstruction of the air filters, which are the most frequently replaced components.

## 3. Problem Description

A train's HVAC system is critical for the comfort of passengers and for proper ventilation. Therefore, a close monitoring of the system is necessary.

The maintainers of the system being studied have developed diagnostics approaches that are based on data. However, these are not fully implemented, because they also use predetermined maintenance on critical components of the system. Predetermined maintenance allows maintainers to establish an interval of time, number of operations, mileage, etc., in order to prevent components from the appearance of failures (EN 13306, 2017). They need to combine PdM and predetermined maintenance to avoid corrective maintenance. This is necessary, because a failure in the HVAC system directly affects people, thus safety becomes more important than the efficiency or reliability of this system. This results in an early substitution of component, ensuring the system meets the required needs of safety, reliability, and effectiveness [1]. However, in transport companies, which use data-driven models, an early replacement of components implies a lack of data on advanced stages of degradation.

The solution is to provide the company with the necessary tools to predict RUL using a hybrid model and, then, to fully deploy PdM while ensuring the system meets the required operational conditions

## 4. Hybrid Model-Based Approaches

The physics-based model of the HVAC system used for generating synthetic data has already been developed, parametrized, and validated in a previous research work [27]. Nevertheless, this research work contains the development of the cabin thermal networking to easy understanding the development.

The physics-based model is used to generate synthetic data, thus intending to overcome the lack of data on advanced stages of degradation, as mentioned above. The development of the physics-based model for building a hybrid model is focused on the improvement of the detectability of FMs by completing the original datasets used to train the data-driven model. Figure 2 shows the expected improvement; the number of FM called CBM in that figure are the number of FMs that the maintainers can currently detect; and the number of FMs defined as HyMA are the FMs detectable by the ongoing development of the HyMA. The FMs that are defined as CBM are extracted from information collected

by the maintainers of the HVAC system. The maintainers use different document, where they have defined the number of FMs that they can detect, the effects of this FMs and their causes, among other information.

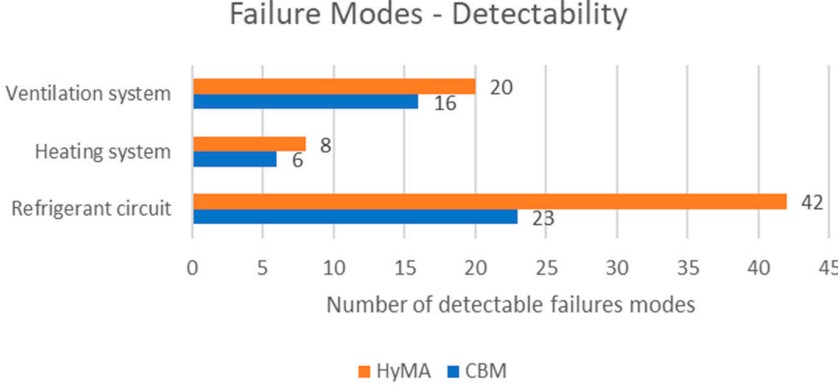

**Figure 2.** Detectability of FMs.

The real system has embedded a few sensors to manage the HVAC system, as listed in Table 1. The definition of a new sensors is necessary to reach the improvement shown in Figure 2. These new sensors modelled are well known as virtual sensors or soft sensors. They are a common tool for fault detection in models based on data, models based on physics and hybrid models [28]. The aim of soft sensors is to deliver additional information for improving diagnostics and prognostics processes. A complete review of soft sensors within the process industry is presented in [29].

**Table 1.** List of sensors used to develop the hybrid model.

| Variable | Type |
|---|---|
| Temperature after compressor 1—virtual | Signal (continuous) |
| Temperature after compressor 2—virtual | Signal (continuous) |
| Temperature before compressor 1—virtual | Signal (continuous) |
| Temperature before compressor 2—virtual | Signal (continuous) |
| Pressure after compressor 1—real | Signal (continuous) |
| Pressure after compressor 2—real | Signal (continuous) |
| Pressure before compressor 1—real | Signal (continuous) |
| Pressure before compressor 2—real | Signal (continuous) |
| Pressure after filter—virtual | Signal (continuous) |
| Pressure before filter—virtual | Signal (continuous) |
| Real heat transfer—virtual | Signal (continuous) |
| $CO_2$ level—real | Signal (continuous) |
| Vehicle temperature—real | Signal (continuous) |
| Impulsion temperature—real | Signal (continuous) |
| Fault code | Condition Variable (discrete) |

The soft sensors that are defined in the physics-based model are related to the measured data. This is considered by utilizing the methodology that is proposed in Figure 3, which was first presented in a previous research work [30] for training, validating, and testing some data-driven models orientated to fault detection.

The data-driven model is trained using the features that were extracted from broth group sensors, real and virtual. Therefore, the measured data must be loaded into the physics-based model to generate the response of the modelled virtual sensors. Moreover, the physics-based model can generate synthetic data in healthy and faulty states by introducing the required inputs. A timeseries of every selected signal is generated after every simulation, the datasets that are related to a simulation are saved in a table, and the simulation is labelled with the level of degradation detected or indicated during the

simulation. The features are extracted from these signals; thus, these features are related to a label and used to build the model; a supervised learning process is then applied. The dataset containing the features related to a simulation is named the "fingerprint".

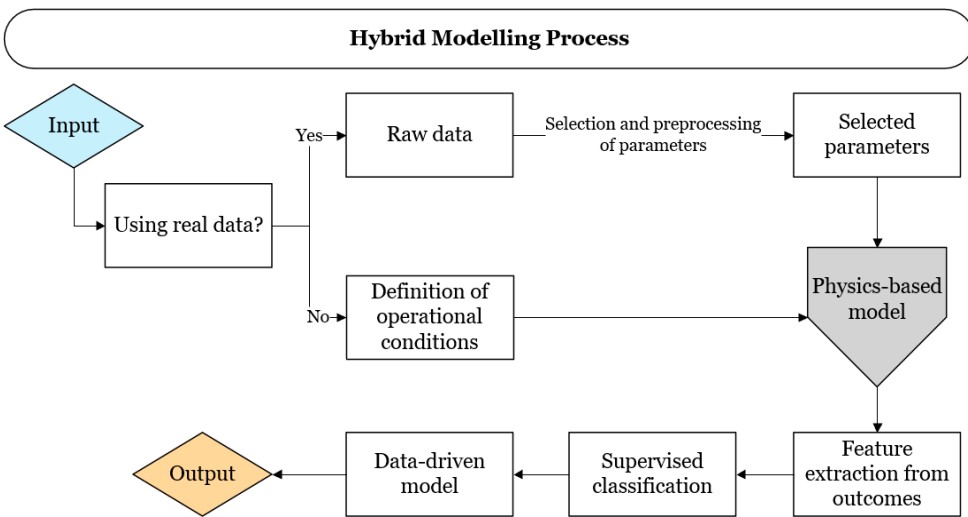

**Figure 3.** Methodology used for building the hybrid model (HyM).

## 5. Physics-Based Model of the HVAC System

The physics-based model of the HVAC is separated into the fresh air demand subsystem, the cabin thermal networking subsystem, and the heating, cooling, and ventilation subsystems. The model is developed and simulated by MATLAB/Simulink version R2019a while using the Simscape toolbox.

Table 1 contains the set of sensors in the real system, which are labelled "real", and the virtual sensors, which are labeled "virtual", as defined in the model. The real sensors manage the system to maintain a comfortable temperature conditions and the required levels of $CO_2$.

The real system does not have a humidity sensor, as shown in Table 1. Therefore, although this parameter is important to comfort, it is not added to the model.

Figure 4 presents the whole HVAC system that was modelled for the cabin. The model contains two cooling subsystems, two heating subsystems, and two ventilation subsystems. The cooling systems share a condenser fan, but not a condenser coil. They are controlled by an automatic switch that manages the temperature inside the cabin.

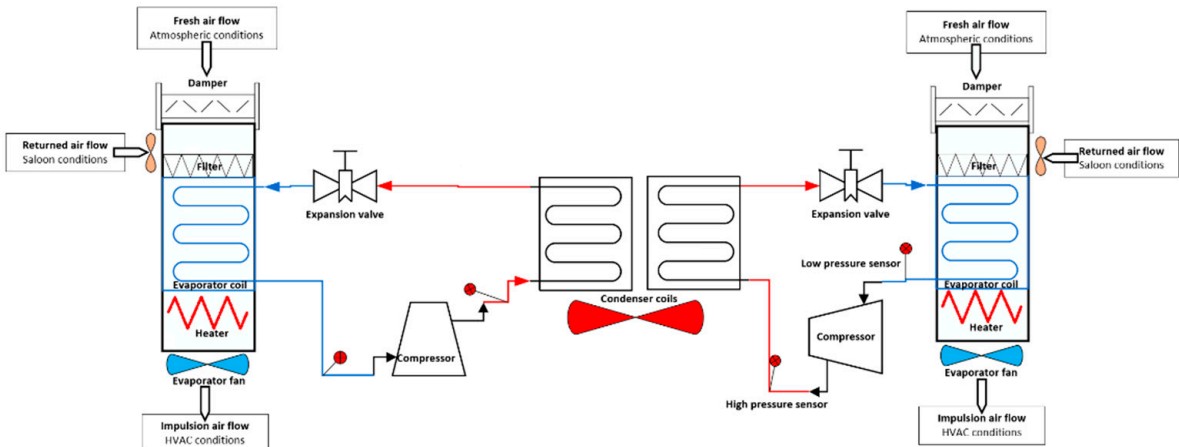

**Figure 4.** Simulink model.

Figure 5 is a block diagram of the cabin thermal networking subsystem showing different heat flows and indicating how the vehicle thermal networking is connected to the HVAC system. The right side of Figure 5 depicts representations of the heat transfer effects from the environment. The heat flows from the subsystems inside the cabin are on the left.

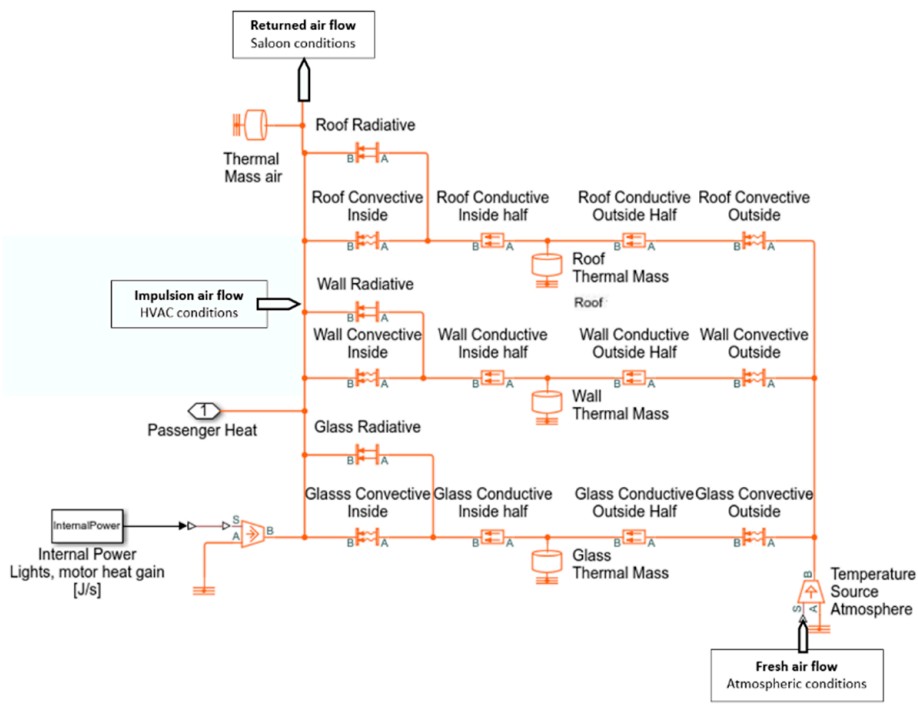

**Figure 5.** Model of the cabin thermal networking.

### 5.1. Fresh Air Demand System

There is a $CO_2$ concentration sensor in the cabin of the vehicle, as shown in Table 1. The output signal of this sensor manages the fresh air damper. It is possible to regulate air flow in four steps based on the $CO_2$ signal. Figure 6 presents how the level of $CO_2$ increases in the cabin, depending on the number of passengers.

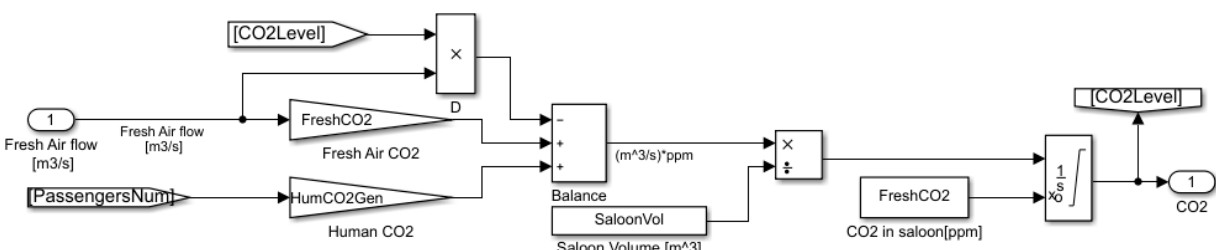

**Figure 6.** Model of the $CO_2$ signal.

This HVAC system was designed following regulation EN 14750-1. The system complies with the requirements of "Climatic Zone III" for summer and "Climatic Zone I" for winter. The fresh air demand subsystem is modelled following Equation (1). It takes the regulation requirements into account, together with the number of passengers and fresh air $CO_2$ concentration.

$$d(C_{saloon})/dt = (C_{pe} \times n + C_{fa} \times \dot{m} - C_0)/V_{saloon} \tag{1}$$

where:

$C_{saloon}$ = $CO_2$ concentration rate of the passenger vehicle (ppm).

$C_{pe}$ = $CO_2$ generation rate per person (ppm/person).
n = number of passengers.
$C_{fa}$ = $CO_2$ concentration rate of fresh air (ppm).
$\dot{m}$ = Mass flow rate (kg/s).
$C_0$ = $CO_2$ concentration rate in the previous integration (ppm).
t = Time.

$C_{saloon}$ indicates the concentration of $CO_2$ in the cabin in particles per million (ppm). This parameter simulates how the concentration of $CO_2$ varies in the cabin. The subsystem opens the fresh air damper when $C_{saloon}$ reaches the maximum value. The fresh air damper remains open until $C_{saloon}$ reaches a satisfactory value.

### 5.2. Cabin Thermal Networking

The physics-based model that is presented in this paper for the thermal networking subsystem keeps the temperature of the cabin within a comfortable range. The model simulates the effects of the HVAC system and various external conditions, including outside temperature, circulation speed of the vehicle, solar radiation, and number of passengers.

The development of the cabin thermal networking model is based on: (1) Fourier's law of heat transfer. This law states that the heat transfer through a material is proportional to the area and the difference in temperatures. This is also known as conduction heat transfer. (2) Newton's law of cooling. This law states that the heat transfer between a heated object and its surroundings is directly proportional to the difference in temperatures between them. This law is also called the convection heat transfer. (3) Stefan–Boltzmann's law. This law establishes that thermal radiation in terms of heat is directly proportional to the fourth power of a black body's temperature.

#### 5.2.1. Heat Transfer from the Environment

The heat transfer from outside the vehicle to inside the cabin is calculated by considering the roof, the walls, and the windows of the vehicle. Heat transfer occurs through forced convection in the side in contact with the external environment. This allows for the speed of the vehicle to be considered. Heat transfer by conduction through the different materials is also considered; this heat transfer is in the form of natural convection and thermal radiation. Figure 7 illustrates the heat transfers that were considered in this paper.

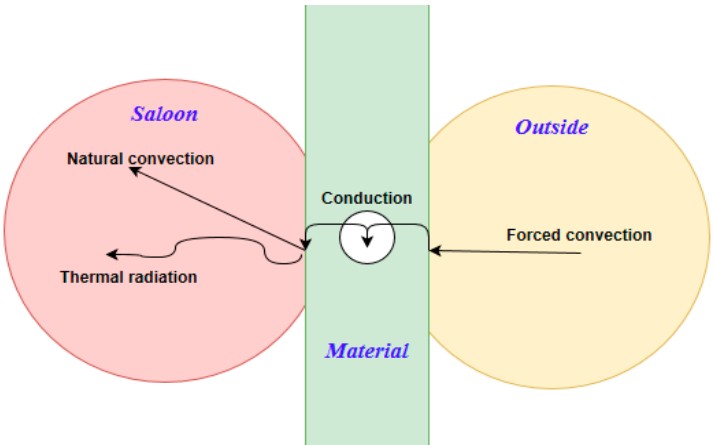

**Figure 7.** Types of heat transfer through materials.

The direct radiation through windows is considered in this analysis and it is calculated while using Equation (2) [31]:

$$Q_{Drad} = I_{solar} \times \alpha \times A \tag{2}$$

where:

$Q_{Drad}$ = Heat transfer in form of thermal radiation (W).
$I_{solar}$ = Solar radiation (W/m$^2$).
$\alpha$ = Absorption coefficient.
A = Area (m$^2$).

Convection Heat Transfer

Heat transfer by convection is defined as the heat transfer from one body to another by the movement of fluids. Two types of convection are introduced in the proposed model: natural convection and forced convection. The former represents the heat transfer by means of fluid movement, which is not generated by any external source. It is calculated by Equation (3), the equation that is used in the Simscape toolbox:

$$Q_{conv} = h \times A \times d(T_2 - T_s)/dt \tag{3}$$

where:

$Q_{conv}$ = Convective heat flow(W).
h = Convection heat transfer coefficient (W/(m$^2$ $\times$ K)).
A = Area in touch with the flux.
$T_2$ = Surface temperature (K).
$T_s$ = Average temperature of the volume (K).
t = Time (s).

Forced convection can also be heat transfer by means of fluid movement, but, in this case, an external source moves the fluids. Here, the heat flux is calculated by Equation (4):

$$Q_{Fconv} = h_F \times L \times d(T_\infty - T_{surf})/dt \tag{4}$$

where:

$Q_{Fconv}$ = Heat transfer by fluid moving over the surface (W).
$h_F$ = Average convection coefficient for laminar flow (J/(m $\times$ kg $\times$ K)).
L = Flow direction (one-dimensional) (m).
$T_\infty$ = Flux temperature out of boundary layers near surface (K).
$T_{surf}$ = Surface temperature in contact with flux on movement (K).
t = Time (s).

The following steps calculate some parameters of Equation (4). These calculations must be completed before the equation can be added to the Simscape toolbox.

Forced Convection

This physics-based model considers forced convection with no phase change occurring within the fluid. However, to determine the appropriate heat flux, the Reynolds number, as in Equation (7), must first be determined to demonstrate the flow is laminar over the surfaces. The Nusselt number, Equation (6), and the last step before using Equation (4), is calculated by the average convection coefficient using Equation (5). Once these equations are developed, they can be adapted to Simscape's convective heat transfer block.

The parameter $h_F$ is defined and calculated by a set of equations see [17]. In this paper, $h_F$ is calculated by Equation (5):

$$h_F = Nu_L \times k \times /L \tag{5}$$

where:

$Nu_L$ = Nusselt number for a laminar fluid
k = Flux specific heat (J/(kg $\times$ K))
L = Flow direction (one-dimensional) (m)

Equation (5) is obtained by making the following approximation of the Nusselt number ($Nu_L$):

$$Nu_L = 0.664 \times Re_L^{1/2} \times Pr^{1/3} \tag{6}$$

where:

$Re_L$ = Reynolds number for laminar fluid.
Pr = Prandtl number. It must be $\geq 0.6$.

This leads to Equation (7):

$$Re_L = u_\infty \times L/\nu \tag{7}$$

where:

$u_\infty$ = Mass flow speed (m/s).
L = Length in the flow direction (one-dimensional) (m).
$\nu$ = Kinetic viscosity.

Thermal Conduction

Thermal conduction is calculated as the heat flow within a body and through the body itself. This physics-based model considers the thermal inertia of windows, walls, and roof. Thermal conduction is calculated in two segments linked by a thermal mass to take the thermal inertia into account.

The thermal conduction is calculated twice by Equation (8):

$$Q_{cond} = 1/2 \times k \times (A/Th) \times d(T_2 - T_1)/dt \tag{8}$$

where:

$Q_{cond}$ = Heat flow by conduction (W).
k = Material thermal conductivity (J/(m $\times$ K)).
A = Surface normal to the heat flow direction (m$^2$).
Th = Thickness of material, distance between surfaces (m).
$T_2 - T_1$ = Temperatures of the surfaces (K).
t = Time (s).

The thermal inertia represents the ability or a combination of abilities of a material to store inertial energy. The thermal inertia is calculated for the different materials by Equation (9):

$$Q_{TInertia} = c \times m \times d(T)/dt \tag{9}$$

where:

$Q_{TInertia}$ = Thermal inertia (W).
c = Specific heat of mass material (J/(kg $\times$ K)).
m = Mass (kg).
T = Temperature (K).
t = Time (s).

Thermal Radiation

Heat transfer by radiation is a consequence of the electromagnetic radiation that is emitted by a body. Radiative heat transfer depends on the body's capacity to emit radiation, its temperature, and the emitting body's surface area. Equation (10) calculates the heat transfer by radiation for the different materials. This equation is also used by Simscape:

$$Q_{Rad} = k_r \times A \times (T_A^4 - T_B^4)/dt \tag{10}$$

where:

$Q_{Rad}$ = Heat flow by radiation (W).
$k_r$ = Radiation coefficient (W/(m$^2$ $\times$ K$^4$)).

A = Emitting body surface area ($m^2$).
$T_A - T_B$ = Temperatures of the materials (K).
t = Time (s).

For the correct use of Equation (10), the radiation coefficient must first be calculated while using Equation (11):

$$k_r = \sigma/((1/\varepsilon_1) + (1/\varepsilon_2) - 1) \tag{11}$$

where:

$\sigma$ = Stefan-Boltzmann constant ($5.67 \times 10^{-8}$ W/($m^2 \times K^4$))
$\varepsilon_1$, $\varepsilon_2$ = Surface emissivity for the emitting and receiving plate, respectively.

### 5.2.2. Heat Transfer from the HVAC System

The heat that is transferred from the HVAC system to the cabin is divided into six parts; two heat flows from the ventilation subsystem, two heat flows from the cooling subsystems, and two heat flows from the heating subsystems.

### 5.2.3. Heat Subsystems

The heating subsystems are developed using Equation (12):

$$Q = \dot{m} \times C_e \times \Delta T/dt \tag{12}$$

where:

Q = Heat transfer in watts (W)
$\dot{m}$ = Mass flow per second (kg/s)
$C_e$ = Specific heat capacity (kJ/(kg $\times$ K))
$\Delta T$ = Temperature difference before and after heat transfer (K)

There are two things of note in the model of the heating system. First, the capacity of the resistors is known. Second, the heat is transferred by the Joule effect, and the heat losses are not directly considered. However, they are evaluated during the parametrization process, where key parameters are adjusted to synchronize the response of the model with the real system.

The temperature is calculated while considering the specific heat capacity of the air, including the mass of the air flow returned from the cabin and fresh air. This means that the temperature is calculated based on the amount of fresh air in the total air volume before heating.

### 5.2.4. Cooling Subsystem

The modelled cooling subsystem is based on the MathWorks Two-Phase Fluid Refrigeration. It includes the evaporator coil and compressor coil and the heat transfer from both fans. This subsystem is defined while using component specifications of the real system using data from the supplier. It is modelled to define the functionalities of the real system in Simulink, indicating how the system works at the points of interest. The real asset only has two sensors to measure the pressure of the refrigerant liquid. These are before and after the compressor. Thus, the model of this part yields important information regarding the cooling subsystem.

### 5.2.5. Ventilation Subsystems

The model contains two ventilation subsystems. It calculates the temperature in the mixed volume of air, i.e., a mixture of fresh air and the returned air from the cabin. Fresh air is taken from outside and it is managed by a damper according to a $CO_2$ signal. Maximum fresh air flow per unit is 1250 $m^3$/h; it can be regulated to a minimum of 25% or closed. Figure 8 represents the air flows that are grouped by temperature.

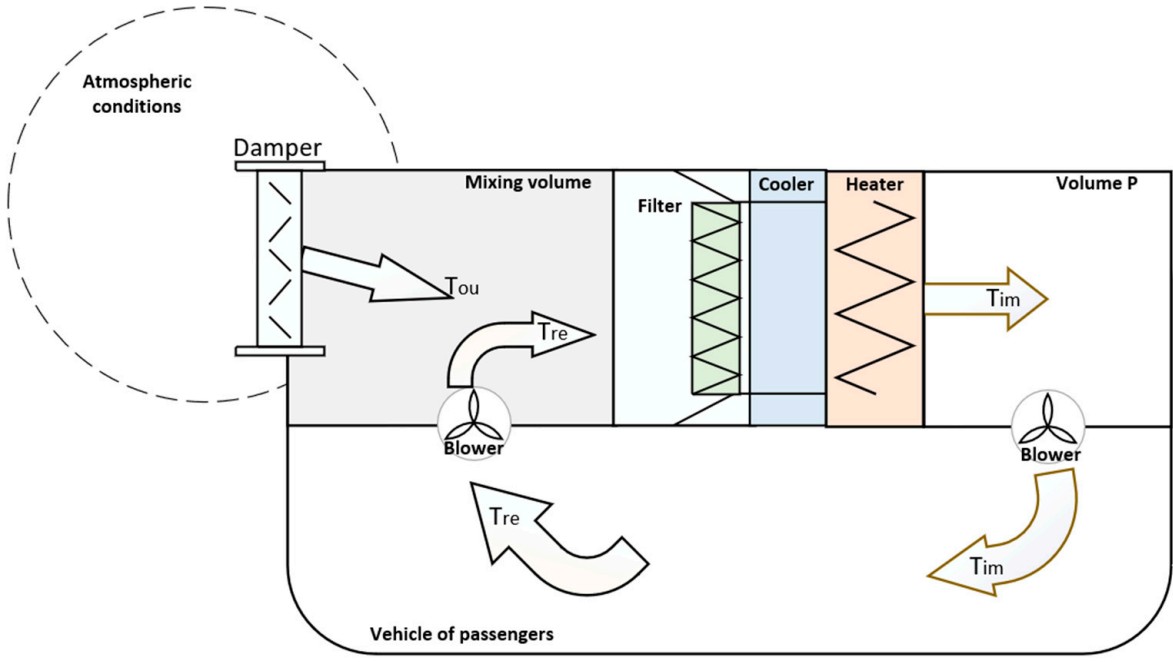

**Figure 8.** Ventilation system, circulation of air flow.

The mixed volume of air presented in Figure 8 includes air from the atmosphere and the vehicle. These two air flows usually have different temperatures, identified as $T_{re}$ and $T_{ou}$. The former is the temperature of the air returned from the vehicle; the latter is the temperature of the air coming from outside. The mix of air flows results in another air flow whose temperature is calculated by Equation (12).

### 5.3. Fault Modelling and Data Generation

The FMs of the HVAC are detected by the sensors that are listed in Table 1. The model presented in this research work includes obstruction of the air filter.

A fault in components is modelled by varying their nominal conditions. Before doing this, it is crucial to evaluate FMEA in order to analyze the FM to be modelled, including its effects and causes. The physics-based model has virtual sensors to improve the detectability of FMs and, then, the prediction of the degradation level of components. Soft sensing is a common tool for improving the detectability of unusual behavior in systems, thus being important when there is an insufficient number of real sensors. In such cases, FMs resulting from different causes may have similar effects in the signals taken from sensors, making it difficult to distinguish between FMs. Hence, the soft sensors are related to a particular FM and to the signals that can be loaded into the model once the hybrid model is implemented. Table 1 contains the virtual sensors that are defined in the physics-based model; these sensors are labelled "virtual".

### 5.3.1. Data Generation

The physics-based model generates synthetic data in healthy and faulty states modelled to train, validate, and test a data-driven model, as aforementioned. The physics-based model has sensors that are located in the real system and soft sensors which generate key features for the detection of faults whose response is inferred by the measured data. Thus, as Figure 3 shows, the data that are collected by sensors embedded in the real system must be loaded into the physics-based model to simulate the response of the soft sensors. The physics-based model can also generate synthetic data in healthy and faulty states by introducing the required inputs. The output of these simulations is recorded in a dataset that contains the data taken from the real system and the data from the soft sensors. Every

simulation is labelled with the mass of dust fed in the air filter. The response of the system is defined by the parameters that are listed in Table 1.

The physics-based model simulates the same response as the real system with the same inputs. Thus, the physics-based model has noise defined in some input parameters. The noise is related to a physical variable and controlled by analyzing the range of values the input can reach.

The data-driven model presented in this research is trained, validated, and tested to estimate the mass of dust in the air filters. The healthy state of the data-driven model is trained, validated, and tested using real data and synthetic data.

### 5.3.2. Feature Extraction

The set of data-driven models developed in this research uses supervised learning methods. The data generated by the physics-based model are organized in a table and labelled with the mass of dust indicated during the simulation.

The HyM presented in this manuscript computes the following statistical features from the signals that are listed in Table 1: mean ($\mu$), standard deviation ($\sigma$), skewness ($\gamma$), kurtosis ($\kappa$), peak value ($x_{peak}$), root mean square (RMS), crest factor (CF), shape factor (SF), impulse factor, and clearance factor.

Mean:
$$\mu = (\Delta t/(t_1 - t_0)) \times \sum\nolimits_{(t=t\_0)}^{t1} \times x(t) \tag{13}$$

Standard deviation (second order moment):
$$\sigma = \sqrt{((\Delta t/(t_1 - t_0)) \times \sum\nolimits_{(t=t\_0)}^{t1} [x(t) - \mu]^2)} \tag{14}$$

Root mean square (RMS):
$$RMS = \sqrt{((\Delta t/(t_1 - t_0)) \times \sum\nolimits_{(t=t\_0)}^{t1} [x(t)]^2)} \tag{15}$$

Shape factor:
$$SF = RMS/(\Delta t/(t_1 - t_0)\sum\nolimits_{(t=t\_0)}^{t1} [x(t)]^2) \tag{16}$$

Skewness (third order moment):
$$\gamma = (\Delta t/(t_1 - t_0)\sum\nolimits_{(t=t\_0)}^{t1} [x(t) - \mu]^3)/\sigma^3 \tag{17}$$

Kurtosis (fourth order moment):
$$\kappa = (\Delta t/(t_1 - t_0)\sum\nolimits_{(t=t\_0)}^{t1} [x(t) - \mu]^4)/\sigma^4 \tag{18}$$

Peak value:
$$x_{peak} = \max |x(t)| \tag{19}$$

Crest factor:
$$CF = x_{peak}/RMS \tag{20}$$

## 6. Data-Driven Model

The learning process uses all of the features extracted from the selected signals. A supervised classification approach is selected to develop the regression model. This machine learning technique develops a function or model able to predict the value of a parameter that is related to a set of features [32]. The regression models compared in this research work include linear regression models, regression trees, Gaussian process regression (GPR) models, support vector machines (SVM), and ensembles of regression trees.

The regression models are trained against overfitting by applying five-fold cross-validation. The results of the models are evaluated in terms of root mean square error (RMSE), which is always positive. The model that obtains the smallest value of RMSE is selected.

Linear regression models use techniques that perform statistical analysis of the relation between two variables by fitting a linear equation to the observed data.

Regression trees models make the prediction by going from observations about an item, which are represented in the branches, to conclusions about the item's target value, which are represented in the leaves. The regression trees models can give continuous values to the target variable [33].

GPR models are nonparametric, i.e., not limited by a functional form; therefore, these models predict the points of interest by computing the probability distribution over all the admissible functions that fit the data [34].

SVM employs linear combinations of different features to make classification decisions. SVM can manage a significant number of features to reach an optimized solution, thus avoiding over-fitting and making feature selection less critical [35].

Ensembles of regression trees are predictive models that are built by a weighted combination of multiple regression trees. This combination of multiple regression trees leads to improving the predictive performance.

## 7. Results and Discussion

A regression model for predicting the mass of solid particles fed in the air filter is developed in his paper. This paper discusses the results obtained after training, validating and testing different regression models. The model is trained using real data and synthetic data generated by the physics-based model.

The prediction model uses GPR, as these obtain the best RMSE, 1.9385, on a validation set. A Bayesian optimization is executed to configure all of the tested models with the hyper-parameter values leading to best predictive performance. The selected technique is tested with isotropic rational quadratic as Kernel function; 284.5569 is the Kernel scale; 0.2359 is the sigma value; the standardize variable is switched to false; and, the basis function is fixed at zero. The RMSE obtained after testing the selected regression model is 1.2376.

The model is trained, validated, and tested using real data and synthetic data. The dataset used during the training and validation processes contains 175 simulations; the dataset used for testing the model contains 52 simulations; and, every simulation defines the behavior of the system with 130 features, which are associated to the mass of solid particles fed in the filter. The performance of the regression model is presented by the RMSE and two plots that indicate the deviations of every predictions.

Figure 9 presents the deviations of the predicted class, and the residuals values associated to these deviations are plotted using the box plot in Figure 10. Equation (21) computes the residual values for each case.

$$\text{Residual} = \text{True value} - \text{predicted value} \qquad (21)$$

Figure 9 contains the grams of solid particles fed in the air filter for the true class, in blue, and for the predicted class in yellow. The wrongly predicted record number are connected to the related true value by a red line, whose length indicates the error.

Figure 10 illustrates the residuals values obtained during validation process. These values are represented in a boxplot that represents the distribution of residuals of every quantity of mass simulated.

Figure 9 shows minor deviations in the model, but, going into further detail, Figure 10 shows how the model is not able to exactly predict the mass of solid particles fed in the air filter, especially at gram levels 49 and 171. Nevertheless, there is not any prediction classified as false negative, which means that any prediction is detected as healthy state, 0 g fed in the air filter, when the true class contained data in faulty state. Moreover, the healthy state does not have huge deviation, there are a few deviations that are shown as outliers in Figure 10, but, apart from the values under 0 g, the deviations are within normal operation values.

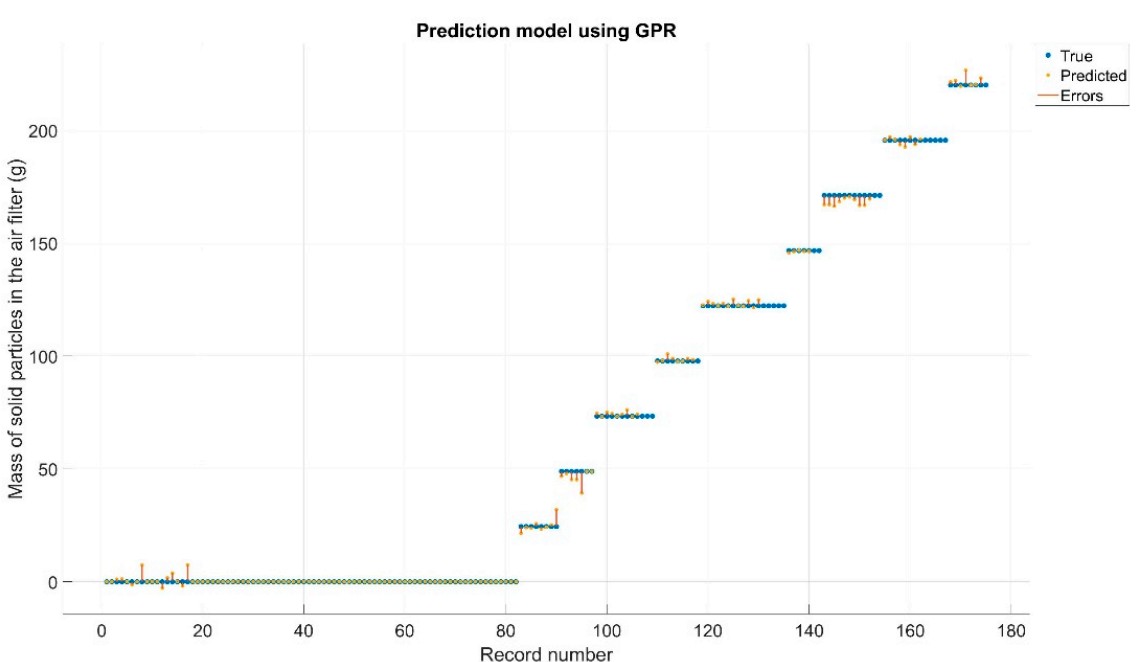

**Figure 9.** Response of the Gaussian process regression model after validation process.

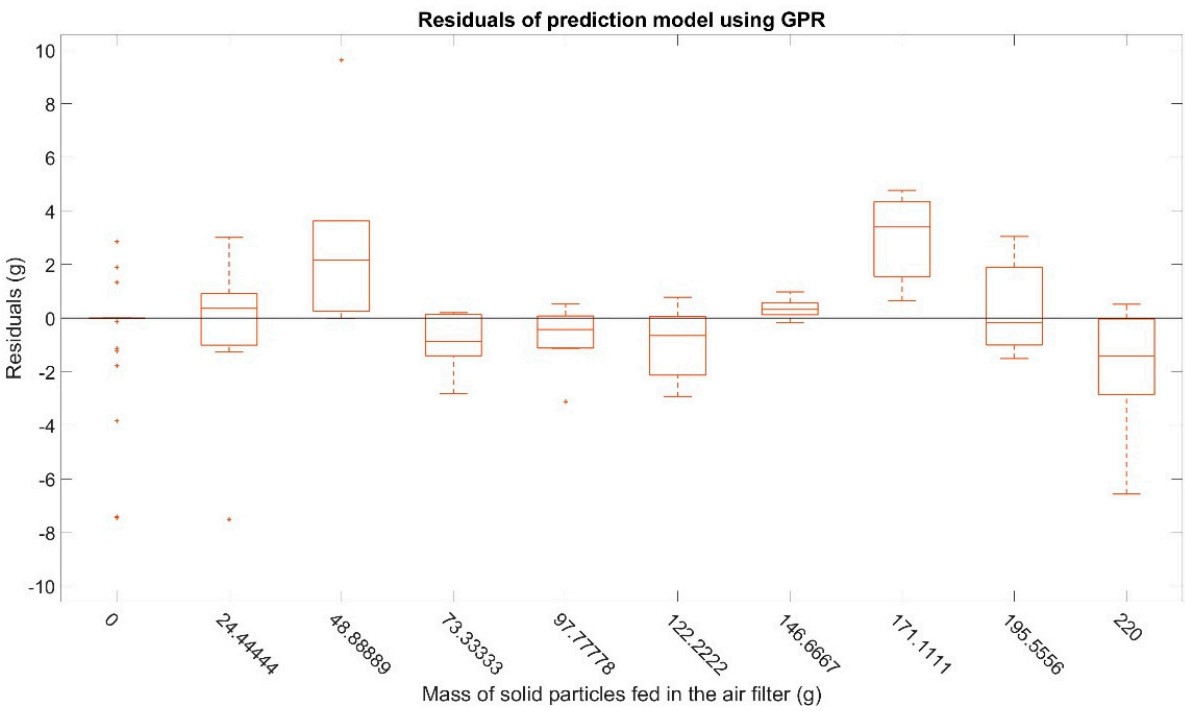

**Figure 10.** Residuals obtained after the validation of the Gaussian process regression model.

## 8. Conclusions and Future Work

The paper is based on the development of a hybrid model for an HVAC system installed in a passenger train carriage. The proposed hybrid model can accurately predict the mass of solid particles that were fed in the HVAC's air filters. A physics-based model, which contains soft sensors, is used to generate synthetic data for the air filter in healthy and faulty states at different levels of degradation. The inputs that are introduced in the physics-based model are real data collected by sensors embedded in the real system. Because the physics-based model generated the same response from the same inputs, noise

is generated and applied to some inputs to increase the diversity of data. This synthetic data and real data are used to train, validate, and test a data-driven model to detect different states of degradation of the air filters.

The proposed hybrid model can accurately estimate the mass of solid particles fed in the air filters. These are the most frequently replaced components, thus implying longer service life, longer time between maintenance tasks, and reduced spare parts inventory, among other things. Definitely, this results in a reduction in the maintenance cost.

The HVAC system is a complex system; it means that several different systems work together to deliver a function. This implies complexity in such aspects as the detection, localization, and identification of faults by the sensors that are embedded in the real system and, then, the prediction of the degradation state of components. The challenge of developing a robust hybrid model for multiple faults increases when it is difficult to acquire faulty data on critical components. Nevertheless, the physics-based model contains key parameters and virtual sensors to improve the detectability of faults. The data generated by the physics-based model can be used for building the HyMA, which must be continuously training the HyMA.

The next steps of this research are related to the development of a hybrid model that is able to detect multiple faults and detect degradation in more components of the HVAC system. This leads to building a multiple fault detection hybrid model and a RUL estimation model. Therefore, these models must be correctly validated and tested using real and synthetic data to ensure that all failures are correctly detected. Finally, the physics-based model will be used to simulate rare conditions to overcome the possibility of "black swans". These analyses will focus on the behavior of the system in new failure modes and the development of methods to identify new and previously undetected failure modes. These futures steps will result in a tool that provides relevant information on the health state of the HVAC system, extends its useful life, reduces its life cycle cost, and improves its reliability and availability.

**Author Contributions:** Conceptualization, D.G., D.S. and A.G.; methodology, A.G.; software, A.G.; validation, A.G., D.S. and D.G.; formal analysis, A.G.; investigation, A.G., D.G. and D.S.; resources, A.G.; data curation, A.G.; writing—original draft preparation, A.G.; writing—review and editing, A.G.; visualization, A.G.; supervision, D.G. and D.S.; project administration, D.G.; funding acquisition, D.G. All authors have read and agreed to the published version of the manuscript.

**Funding:** This research was funded by the Basque Government, through ELKARTEK (ref. KK-2020/00049) funding grant.

**Institutional Review Board Statement:** Not applicable.

**Informed Consent Statement:** Not applicable.

**Data Availability Statement:** Not applicable.

**Conflicts of Interest:** The authors declare no conflict of interest.

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
