# Peer review of "Hybrid Model Development for HVAC System in Transportation"

_technologies, doi:10.3390/technologies9010018_

Round 1

Reviewer 1 Report

This manuscript with the title “Hybrid Model Development for HVAC System in Transportation” discusses the use of a physics-based Simulink model for modelling HVAC. The technical work can be considered for this journal; however, some claims are not accurate. It also seems that authors are forcefully claiming that the work is somehow related to “data-driven” modelling when it is just a parameterized Simulink model. My comments are as following:

  1. Using Simulink to model HVAC is not novel (see a few of the many papers below). Authors did not clearly specify the novelty of this work. It is not known what the contribution of this work for this research direction is. This should be covered in the last paragraph of the introduction.

https://doi.org/10.1016/j.enbuild.2013.09.016

https://doi.org/10.1191/0143624403bt079oa

https://doi.org/10.1016/j.buildenv.2007.11.001

  1. It is not clear where does figure 3 come from? Authors should explain this in the paper.
  2. Section 2.1 “Advantages of hybrid model-based approaches” is not a problem description, so it should not be included in section 2.
  3. Authors argued the importance of data-driven, experience-based and model-based approaches in the introduction and section 2.1. However, this paper only uses a physics-based model which was parameterized. There are no data-driven methods in this paper at all. Although authors claim that this physics-based model is a part of the hybrid model, the other part of the hybrid model was never presented. Authors also did not clearly present how the hybrid model integrates these two parts.
  4. It is also not very logical that the model is a realistic representation of an HVAC system if the whole system is first-ordered and there are no delays/dead-time. Is there any way to validate the results?
  5. Again, in the conclusion, the authors started the paragraph with “Data-driven approaches have good results in the maintenance of the studied HVAC system” but this paper does not contain any data-driven approaches. Authors should address what is meant by “data-driven approaches”.
  6. The cited works are too lacking for this topic. There are many works that have used Simulink to model HVAC systems. Authors should cite more of them and give credit to the previous works that were already published. I reckon that the total citations should be more than 30.

Author Response

Dear reviewer, 

Please, see the attachment, there is described how I applied all your comments to the paper. All your comments in the updated version of the paper.

Thank you so much for your comments, they are extremely useful for improving this paper and future paper.

Best regards,

Antonio Gálvez

Reviewer 2 Report

In this study, physics-based model has been proposed for a Heating, Ventilation, Air Conditioning (HVAC) system installed in the passenger vehicle of a train. The physics-based model is divided into four main parts: heating subsystems, cooling subsystems, ventilation subsystems, and cabin thermal networking subsystems. The proposed physical model of an HVAC system is developed using MATLAB R2019b. In my opinion, the quality of the manuscript is not meeting the standards of “Technologies” in the current form and should be revised significantly. The following reasons justify the recommendations:

  • It appears that manuscript is prepared casually, figures 9-10 in the results and discussion section do not even includes axis titles in the graphs. Although, the information for the Y-axis title is given in the text but the information of X-axis title is missing. It is hard to comment on the results based on the presented graphs.
  • The central theme of this paper is development of hybrid-based model. However, nowhere in the text it is clear how physics-based model is clubbed with the data driven model to get a hybrid-based model. Section-2 briefly discussed about the failure mode detection by hybrid-based model and condition-based monitoring. An elaborative discussion should be added on how hybrid-based model was developed and on what basis it was compared with the CBM i.e how the inputs for CBM were collected, what was the size of data set, what is data driven model and how it was coupled with the physics-based model?
  • The proposed physics-based model is a simplified model. Its prediction reliability should be examined, and a validation study should be added before counting on the results.
  • Authors added a significant discussion on the prognostic and the diagnostic maintenance and on the remaining useful life but did not mention how the hybrid-based model or physics-based model was used to predict RUL. This should be explained.

This manuscript needs significant correction in the quality of all the figures.

Author Response

Dear reviewer, 

Please, see the attachment, you can find described how I applied all your comments to the paper. All your comments are considered in the updated version of the paper.

Thank you so much for your comments, they are extremely useful for improving this paper and future paper.

Best regards,

Antonio Gálvez

Reviewer 3 Report

The paper is about the hybrid model development in transportation. The area of research is an exciting area for further research. The authors found a significant research gap. However, the paper must be revised based on the following comments. 1. As it is a hybrid system, it has to compare with both those systems to show the hybrid case's improvement. 2. The abstract should not contain the abbreviation. The significant findings and novelty of this study must be indicated in the abstract. 3. The introduction and literature review should be separated. 4. Bulky references should not be added, and the introduction should contain the exact research gap and orientation of the paper (Please add those references for more reader-friendly of the article https://doi.org/10.3390/math8030357 and https://doi.org/10.3390/en12142789). 5. Do not use we/our/us throughout the study. Replace those sentences in other ways. 6. The validity of data should be provided. 7. The source of numerical data should be mentioned. 8. Managerial insight should be mentioned. 9. The proper extension of this paper with proper references should be added at the end of the conclusions.

Author Response

Dear reviewer, 

Please, see the attachment, you can find described how I applied your comments to the paper. All your comments are considered in the updated version of the paper.

Thank you so much for your comments, they are extremely useful for improving this paper and future paper.

Best regards,

Antonio Gálvez

Round 2

Reviewer 2 Report

The manuscript has been modified and can be accepted for publications.